# Intrinsic Expression of Coagulation Factors and Protease Activated Receptor 1 (PAR1) in Photoreceptors and Inner Retinal Layers

**DOI:** 10.3390/ijms23020984

**Published:** 2022-01-17

**Authors:** Zehavit Goldberg, Ifat Sher, Lamis Qassim, Joab Chapman, Ygal Rotenstreich, Efrat Shavit-Stein

**Affiliations:** 1Goldschleger Eye Institute, Sheba Medical Center, Ramat Gan 5266202, Israel; Golberg.Zeahavit@sheba.health.gov.il (Z.G.); Ifat.SherRosenthal@sheba.health.gov.il (I.S.); Ygal.Rotenstreich@sheba.health.gov.il (Y.R.); 2Sackler Faculty of Medicine, Tel Aviv University, Tel Aviv 6997801, Israel; 3Department of Neurology, Sheba Medical Center, Ramat Gan 5266202, Israel; Lamis.Qassim@sheba.health.gov.il (L.Q.); joab.chapman@sheba.health.gov.il (J.C.); 4Department of Neurology and Neurosurgery, Sackler Faculty of Medicine, Tel Aviv University, Tel Aviv 6997801, Israel; 5Department of Physiology and Pharmacology, Sackler Faculty of Medicine, Tel Aviv University, Tel Aviv 6997801, Israel; 6Robert and Martha Harden Chair in Mental and Neurological Diseases, Sackler Faculty of Medicine, Tel Aviv University, Tel Aviv 6997801, Israel; 7Sagol School of Neuroscience, Tel Aviv University, Tel Aviv 6997801, Israel

**Keywords:** thrombin, PAR1, neuroretina, rods, cones, photoreceptor

## Abstract

The aim of this study was to characterize the distribution of the thrombin receptor, protease activated receptor 1 (PAR1), in the neuroretina. Neuroretina samples of wild-type C57BL/6J and PAR1^−/−^ mice were processed for indirect immunofluorescence and Western blot analysis. Reverse transcription quantitative real-time PCR (RT-qPCR) was used to determine mRNA expression of coagulation Factor X (FX), prothrombin (PT), and PAR1 in the isolated neuroretina. Thrombin activity following KCl depolarization was assessed in mouse neuroretinas ex vivo. PAR1 staining was observed in the retinal ganglion cells, inner nuclear layer cells, and photoreceptors in mouse retinal cross sections by indirect immunofluorescence. PAR1 co-localized with rhodopsin in rod outer segments but was not expressed in cone outer segments. Western blot analysis confirmed PAR1 expression in the neuroretina. Factor X, prothrombin, and PAR1 mRNA expression was detected in isolated neuroretinas. Thrombin activity was elevated by nearly four-fold in mouse neuroretinas following KCl depolarization (0.012 vs. 0.044 mu/mL, *p* = 0.0497). The intrinsic expression of coagulation factors in the isolated neuroretina together with a functional increase in thrombin activity following KCl depolarization may suggest a role for the PAR1/thrombin pathway in retinal function.

## 1. Introduction

The thrombin protease activated receptor 1 (PAR1) is a G-protein-coupled receptor and one of four members of protease-activated receptors (PARs) [1]. Thrombin, a serine protease, cleaves PAR1 at its extracellular N-terminus domain, exposing a PAR1 self-ligand domain [2].

PAR1 and its main activator thrombin were found to play key roles in pathologies of the central and peripheral nervous systems, including in Parkinson’s disease, amyotrophic lateral sclerosis (ALS), glioblastoma (GBM), diabetic neuropathy (DN), and sciatic nerve injury [3,4,5,6,7]. PAR1 is expressed in the sciatic node of Ranvier, where its activation leads to conduction block [8], and its increased activation on Schwann cells following injury was suggested to limit axonal repair [9]. Therefore, the PAR1/thrombin pathway has been marked as a potential target for the treatment of these diseases.

Several studies indicate that PAR1 is expressed in retinal pigment epithelial (RPE) cells, pointing to a possible role in the integrity of the retinal–blood barrier (RBB). Exposure of cultured RPE cells to thrombin results in the formation of intercellular gaps [10]. In addition, thrombin induces upregulation of vascular endothelial growth factor (VEGF) in human retinal microvascular endothelial and RPE cells [11] and increases the secretion of vascular endothelial growth factor (VEGF) in vivo [12]. Studies in diabetic rats indicated that increased VEGF levels in the retina are associated with retinal vascular alterations [13]. Thrombin via PAR1 was found to facilitate angiogenesis and induce human endothelial RPE cell migration in culture [12,13,14]. In addition, in a study using a human RPE cell line, thrombin induced the expression of inflammatory cytokines and growth factors such as tumor necrosis factor α (TNF-α) and interleukin 1β (IL-1β), IL-6, and IL-8 in a PAR1 dependent manner [15,16]. IL-1β was found to increase cellular apoptosis in bovine retinal endothelial cells in vitro [16]. Together, these studies support the role of thrombin/PAR1 and VEGF in angiogenesis and RBB breakdown in proliferative retinopathies [11,12,13,14].

Recent studies suggested that PAR1 and thrombin are expressed within the ocular microenvironment of patients with proliferative diabetic retinopathy (PDR), in epiretinal membrane vascular endothelial and stromal cells, as well as in intravascular leukocytes. High thrombin levels were found in the vitreous body of PDR patients [11]. Together with the above indicated in vitro studies, it was suggested that the PAR1/thrombin pathway contributes to PDR progression [11,12].

Until today, the description of PAR1 expression in the neuroretina has been limited to the retinal ganglion cells (RGC) and Müller glial cells [17,18] where it was reported to mediate calcium mobilization. The expression of PAR1 in photoreceptors and the inner nuclear cell layer has not been reported before. Here, we describe, for the first time, that PAR1 is expressed in the outer neuroretina in rod photoreceptors but not in cones, as well as in the inner nuclear layer. We demonstrate an elevation in thrombin activity in isolated neuroretinas in response to KCl depolarization ex vivo.

## 2. Results

### 2.1. PAR1 Expression Pattern in Mouse Neuroretina by Immunofluorescence

To determine the distribution of PAR1 protein in mouse neuroretina, eyes were enucleated from ten C57BL/6J male mice and were paraffin embedded, and retinal sections were stained with an anti-PAR1 antibody. PAR1 was detected in the ganglion cell layer, inner nuclear layer, and outer nuclear layer (Figure 1A,C), suggesting that the majority of the neuroretinal cells express PAR1. No staining was observed in neuroretinas derived from PAR1^−/−^ mice (Figure 1D) nor in retinal sections that were incubated only with the secondary antibody (Figure 1E) or with commercial mouse IgG control antibody (Figure 1F). As previous studies suggested that PAR1 is expressed in retinal ganglion cells in vitro [17], the staining of these cells served as an internal positive control (Figure 1A,C).

To further characterize PAR1 expression patterns in retinal photoreceptors, retinas were co-stained with PAR1 and antibodies specifically targeting the rod and cone photoreceptor cells (anti-rhodopsin and anti M/L or S-opsin, respectively). As shown in Figure 2, PAR1 co-localized with rhodopsin in rod outer segments. Pearson’s correlation coefficient between the PAR1 (red channel) and rhodopsin (green channel) was 0.92 ± 0.02 (Figure 2E,F), indicating a strong overlap between PAR1 and rhodopsin. By contrast, a weak, almost undetectable PAR1 staining was observed in M/L and S-cone outer segments as indicated by co-staining with M/L and S-opsin antibodies, respectively (Figure 3). Colocalization analysis showed a low correlation between PAR1 and M/L- and S- opsin staining with Pearson correlation coefficients of 0.11 ± 0.110 and 0.04 ± 0.01, respectively.

### 2.2. PAR1 Expression in the Neuroretina by Western Blot Analysis

To confirm antibody staining specificity and further assess the expression of PAR1 proteins in the neuroretina, Western blot analysis was performed using the same anti-PAR1 antibody used for the immunofluorescence analysis. This antibody (NBP-71770, Novus biologicals) is directed against amino acid sequence Asp35–Arg46, which includes the thrombin cleavage site of PAR1 (Arg41) and the six preceding amino acids. Hence, this antibody is predicted to mainly detect the intact PAR1 protein (~52 kDa). Neuroretinas collected from PAR1^−/−^ mice served as a negative control for the Western blot analysis. As previous studies indicated that PAR1 is not expressed in mouse platelets [19], lysates of mouse platelets were used as an additional negative control for the analysis.

Western blot analysis, with the anti-PAR1 antibody, demonstrated the expression of PAR1 in the mouse neuroretina, as demonstrated by a specific band of ~52 kDa recognized by this antibody. The 52 kDa band was not detected in lysates of mouse platelets or PAR1^−/−^ neuroretinas (Figure 4).

### 2.3. Expression of PAR1, Coagulation Factor X, and Prothrombin mRNA in Mouse Neuroretina

To determine the source of the coagulation proteins and PAR1 in the neuroretina, qRT-PCR analysis was performed on isolated neuroretinas (n = 6 mice). Isolated neuroretinas contain retinal neurons and glial cells but lack RPE cells and choroid. The qRT-PCR analysis indicated the expression of PAR1, coagulation Factor X (FX) and prothrombin mRNA in the isolated neuroretina (Figure 5). The mRNA expression level of PAR1 was higher by two orders of magnitude in comparison to FX and prothrombin (0.0202 ± 0.0022, 0.0002 ± 0.0000, 0.0002 ± 0.0000, normalized to the house keeping gene hypoxanthine guanine phosphoribosyl transferase (HPRT) expression, *p* < 0.0005).

### 2.4. Thrombin Activity Is Elevated in Isolated Neuroretinas Ex Vivo following KCl Depolarization

KCl depolarization is a well-established method for depolarization of isolated neuroretina ex vivo [20,21,22,23,24]. To test the possible functional role of the thrombin/PAR1 pathway in the neuroretina, thrombin activity was measured ex vivo in neuroretinas under KCl depolarizing conditions (56 mM KCl, “high KCl”) in comparison to low KCl conditions (5.6 mM KCl).

Thrombin activity was increased nearly four-fold in the presence of high KCl concentrations in comparison to low KCl concentrations (0.044 ± 0.013 vs. 0.013 ± 0.005 mu/mL, *p* = 0.0497, Figure 6).

## 3. Discussion

The present study demonstrates, to the best of our knowledge and for the first time, that PAR1 is expressed in mouse inner and outer neuroretinal layers. The expression of PAR1, prothrombin and FX mRNA in isolated neuroretinas supports the local production of coagulation proteins in the neuroretina. A functional role of this pathway is suggested by the KCl depolarization experiment, indicating increased thrombin activity in the neuroretinal following depolarization. Together, these findings are the first evidence that the thrombin–PAR1 pathway may play a role in neural function in the retina.

Our qRT-PCR data indicate the expression of PAR1, FX and prothrombin mRNA in the neuroretinas of mature (13-week-old, P91) mice, and support the intrinsic production of these coagulation factors in neuroretinal cells. PAR1 mRNA levels were 100-fold higher in the neuroretina compared to prothrombin and FX. These findings are in-line with published single-cell analyses of mouse developing retina (E11-P8), indicating a significant higher number of mouse retinal cells expressing PAR1 compared to FX and prothrombin (one and two magnitudes of order higher, respectively) [25]. These findings may be explained by the fact that FX is the initiator and a limiting factor in the coagulation cascade. Thus, low levels of FX expression in the neuroretina may tightly regulate the activation of thrombin-PAR1 located downstream in this signaling cascade.

PAR1 protein was co-localized with rhodopsin, the light absorbing protein in rod photoreceptors, but not with the M/L- or S- opsin, which mediate light absorption in M/L- and S-cones, respectively. In mice, similar to primates, rods constitute the vast majority (nearly 97%) of photoreceptors, and cones account for the remainder [26]. As rods are more sensitive to light than cones, mediating light detection at a single-photon level [27], our findings may suggest a possible role for PAR1 in visual function under low light conditions [28].

We have tested four different commercial antibodies directed against PAR1 (Appendix A). As PAR1 was shown to be expressed in cell cultures of RGCs and RPE cells in vitro [10,17], immunofluorescent staining of these cells was used as a positive control for antibody staining in the retinal sections. Only the Novus antibody stained RGC and RPE cells. Hence this antibody was used in this study. FASTA on NCBI BLAST analysis ruled out possible non-specific binding of the antibody. The specificity of this antibody was further demonstrated by detection of the protein at the expected molecular weight by Western blot analysis and lack of protein detection in neuroretinal lysates of PAR^−/−^ mice or mouse platelets.

Manipulating the KCl extracellular concentration was suggested to lead to acetylcholine release from amacrine and/or bipolar cells in the inner retina [20]. Our data revealed significant elevation in thrombin activity in the neuroretina during KCl depolarization, suggesting that thrombin may have a role as a neuromodulator in the neuroretina. Our findings are in line with previous studies indicating the involvement of the thrombin/PAR1 pathway in retinal Müller cells and in neuronal activity in the brain and the sciatic nerve [8,18]. Thus, exposure of a human retinal Müller cell line in vitro to thrombin inhibited the potassium channel Kir4.1 that maintains the extremely negative resting membrane potential in Müller cell and plays a key role in glutamate balance and neuronal activity in the retina [18,29,30,31,32,33,34,35,36]. In addition, the PAR1/thrombin pathway was shown to regulate glutamate-dependent long-term potentiation (LTP) in mice hippocampal slices [37]. In the peripheral nervous system, exposing the rat sciatic nerve to high thrombin levels led to a nerve conduction block [8], and increased PAR1 activation following nerve injury was suggested to limit axonal regeneration [9]. The role of PAR1 in retinal function in vivo remains to be determined.

One of the study limitations is that the mRNA expression analysis was conducted on non-perfused neuro-retinas. Endothelial cells constitute only ~0.6% of all retinal cells [38] and the choroid was removed before tissue lysis. Since mice platelets lack PAR1 expression [19], we hypothesize that other blood cells and retinal endothelial cells are negligible relative to all neuroretinal cells.

Taken together, our data demonstrate the intrinsic and specific co-localization of PAR1 and rhodopsin in rod photoreceptor outer segments and cell bodies, as well as inner nuclear layers of the retina, and identify thrombin activity during KCl-induced retinal depolarization. Together with the elevated thrombin activity detected following ex vivo depolarization, our findings suggest a possible involvement of the thrombin/PAR1 pathway in neuroretinal function.

## 4. Conclusions

The coagulation pathway proteins FX, prothrombin, PAR1 and thrombin are expressed in the neuroretina, with PAR1 specifically expressed in rod, but not in cone, photoreceptors, and intrinsic thrombin activity is affected by KCl depolarization. Hence, the PAR1/thrombin pathway may play a role in retinal function, specifically in night vision, and this may have consequences in neuroretinal diseases.

## 5. Materials and Methods

### 5.1. Animals

The study was carried out with 13-week-old male C57BL/6JOlaHsd mice (purchased from Envigo Laboratories, Rehovot, Israel) and male PAR1 KO C57BL/6J background mice (PAR1^−/−^, a generous gift of Prof. Yair Reisner, the Weizmann Institute of Science, Rehovot, Israel). Mice were housed at the Sheba Medical Center animal facility. All animal procedures and experiments were approved by the Sheba Medical Center Institutional Animal Care Committee (1210/19-ANIM, 8 August 2019) and conformed to recommendations of the Association for Research in Vision and Ophthalmology Statement for the Use of Animals in Ophthalmic and Vision Research and according to the ARRIVE (Animal Research: Reporting of In Vivo Experiments) guidelines.

### 5.2. Immunofluorescence Staining

Mice were sacrificed at 13 weeks of age, and their eyes were fixed in 4% formaldehyde, as previously described [39,40]. Retinal paraffin sections were deparaffinized and rehydrated. Following epitope retrieval using citrate buffer (pH 6.0, Zytomed Systems GMBH, Berlin, Germany), retinal sections were blocked with 10% donkey serum in phosphate buffered saline (PBS) containing 0.1% Triton X-100 followed by incubation with primary antibodies (diluted in PBS containing 0.1% Triton X-100 and 1% donkey serum) as detailed in Appendix A. Next, sections were extensively washed in PBS and incubated with fluorescently labeled secondary antibodies as detailed in Appendix A. Negative controls were performed by omitting primary antibodies and staining only with fluorescence secondary antibodies, by using equivalent concentration of commercial mouse whole IgG (Jackson Immune Research, West Grove, PA, USA) and by using retinal sections derived from PAR1^−/−^ mice. Sections were mounted with aqueous mounting medium containing 4′,6-diamidino-2-phenylindole (DAPI Fluoromount-G, Eelectron Microscopy Sciences, Hatfield, PA, USA), and viewed with a confocal microscope (Confocal Microscope, ZEISS, LSM 700).

### 5.3. Co-Localization Analysis

Co-localization analysis was performed utilizing Image Zen software (LSM software ZEN 2012 SP1 Version 8.1, Zeiss, Jena, Germany) on three separate regions of interest (ROI) in retinal sections of two mice. Results are presented as Pearson’s correlation coefficients of merged fluorescence histograms after selecting a uniform ROI for each protein.

### 5.4. Isolation of Mouse Neuroretinas

Mice were euthanized by IP injection of pentobarbital, 100 µL (Pentobarbital 20%, CTS, Kiryat Malachi, Israel). Eyes were removed and placed in ice-cold PBS. Corneas were removed by punching a hole in the limbus using a 25-G needle, followed by an incision around the periphery of the cornea. The lens was removed, and the neuroretina was separated from the RPE and the sclera using two pairs of blunt tip tweezers. The procedure was performed on ice under a stereo microscope (SMZ745T, Nikon Instruments, Melville, NY, USA).

### 5.5. Protein Extraction from Mouse Neuroretinas and Platelets

Mouse neuroretinas were isolated as indicated above and homogenized in radioimmunoprecipitation assay (RIPA) buffer (50 mM Tris HCl, pH 7.6, 150 mM NaCl, 1% NP-40, 0.5% Sodium Deoxycholate, and 0.1% SDS supplemented with commercial Protease Inhibitor Cocktail (P-2714, Sigma-Aldrich, Saint Louis, MO, USA)) using a bead-based homogenizer (BB*24B, Next Advance, Troy, NY, USA). For platelet preparation, blood was collected in tubes containing CPDA1 (citric acid, sodium citrate, monobasic sodium phosphate, dextrose, and adenine). Blood was centrifuged (100× *g*, 9 min) to separate the platelets rich plasma (PRP). Residual erythrocytes were removed by centrifugation (100× *g*, 6 min). The PRP was centrifugated (1000× *g*, 5 min) to sediment the platelets, and the supernatant was removed. Platelets were resuspended in PBS and were centrifuged (1000× *g*, 3 min) to wash the platelets and remove residual plasma. Platelets were resuspended in PBS for protein concentration determination and further Western blot analysis. A bicinchoninic acid (BCA) kit (QPRO-BCA kit, PRTD1,0500, Cyanagen, Bologna, Italy) was used to determine protein concentration.

### 5.6. Western Blot Analysis

Twenty micrograms of total protein samples from the neuroretinas and five micrograms of total proteins from platelets were separated by polyacrylamide gel electrophoresis and transferred onto nitrocellulose membranes for Western blot analysis. Membranes were incubated overnight (16 hour) at 4 °C with primary mouse anti-PAR1 antibody (1:500, NBP-71770, Novus biologicals) in Tris-buffered saline (50 mM Tris HCl pH 7.6, 150 mM NaCl, 0.1% Tween 20). Membranes were washed and incubated at room temperature with horseradish peroxidase-conjugated goat anti-mouse secondary antibodies (Jackson Immuno Research Laboratories, West Grove, PA, USA). Enhanced chemiluminescence (ECL) method (MYECL Imager, Thermo Scientific, Waltham, MA, USA) was used for protein detection.

### 5.7. Quantitative Reverse Transcription Real-Time PCR (qRT-PCR)

Animals were euthanized with pentobarbital. Neuroretinas were isolated as indicated above and were homogenized with a bullet blender homogenizer (BB*24B, Next Advance, Troy, NY, USA) at a maximum speed for two minutes. RNA was extracted from the neuroretina using RNA, Aurum™ Total RNA Mini Kit (Bio Rad Laboratories, 7326820 Hercules, CA, USA) following manufacturer’s instructions. One microgram of total RNA was used for reverse transcription using high-capacity cDNA reverse transcription kit (Applied Biosystems, AB-4374966, Thermo Fisher Scientific, Waltham, MA, USA). The qRT-PCR was performed on the StepOne™ Real-Time PCR System (Applied Biosystems, Rhenium, Modiin Maccabim Reut, Israel) using Fast SYBR Green Master (Applied Biosystems, AB-4385612, Thermo Fisher Scientific, Waltham, MA, USA). A standard amplification program was used, 1 cycle of 95 °C for 20 s, 40 cycles of 95 °C for 3 s and 60 °C for 30 s. The results were normalized to a reference gene expression, hypoxanthine guanine phosphoribosyl transferase (HPRT) within the same cDNA sample and calculated using the 2^ΔCT method. Results are presented as fold change relative to HPRT and reported as mean ± standard error (SE). The primers used in the qRT-PCR analysis are listed in Appendix A.

### 5.8. Thrombin Activity Assay

Neuroretinas were isolated as described and were divided into two halves, by cutting the retina symmetrically through the optic nerve under dim red-light conditions that were kept throughout the experiment. Each half was placed into a single well in 96-well black microplates (Nunc, Roskilde, Denmark) containing a thrombin substrate buffer (in mM: 150 NaCl, 1 CaCl_2_, 50 Tris-HCl, pH 8.0), bovine serum albumin (BSA, 0.1%, 9048-46-8, Amresco, Solon, OH, USA), bestatin (0.1 mg/mL, 70520, Cayman-Chemical Company, Ann Arbor, MI, USA) and prolylendopeptidase inhibitor (0.2 mM, 537011, Calbiochem, San Diego, CA, USA). Thrombin enzymatic activity was measured using a fluorometric assay, based on the cleavage rate of the synthetic thrombin fluorogenic substrate Boc-Asp (OBzl)-Pro-Arg-AMC (14 µM, I-1560, Bachem, Bubendorf, Switzerland) as previously described [7]. Briefly, the cleavage rate of the substrate was defined by the linear slope of the fluorescence intensity versus time [41,42,43]. First, the fluorescence measurement was conducted in thrombin activity buffer [42] every two minutes four times to establish the baseline activity of thrombin in the neuroretina. KCl depolarization of the neuroretina was achieved by addition of KCl to a final concentration of 56 mM to one half of each neuroretina (n = 10). The second half of the neuroretina was incubated in a low (5.6 mM) KCl buffer (n = 10). The fluorescence signal was measured by a microplate reader (Infinite 2000; Tecan, Männedorf, Switzerland) with excitation and emission filters of 360 ± 35 nm and 460 ± 35 nm, respectively. A calibration curve was used in each experiment with 0.00078–0.05 u/mL bovine thrombin (T-4648, Sigma-Aldrich). The neuroretinal halves were weighed following the thrombin activity assay. No significant differences were obtained between the weights of neuroretinas incubated in low and high KCl (*p* = 0.88).

### 5.9. Statistics

Pearson’s correlation coefficients of co-localization analysis was performed using Image Zen software (LSM software ZEN 2012 SP1 Version 8.1, Zeiss, Jena, Germany). Thrombin activity results were analyzed by paired, two-tailed t-test analyses. All statistical analyses were performed using GraphPad Prism 8 (GraphPad Software Inc., San Diego, CA, USA). Group size was determined based on previous studies [40,44].

## Figures and Tables

**Figure 1 ijms-23-00984-f001:**
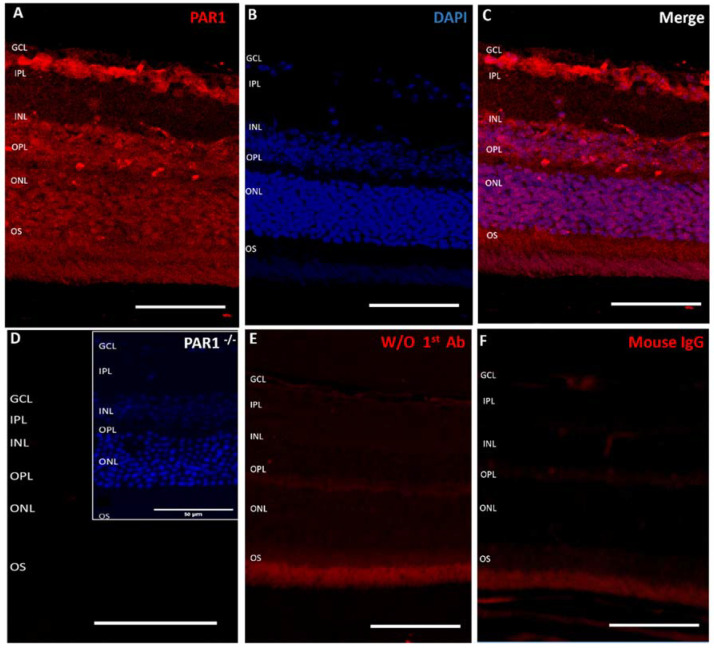
PAR1 expression in mouse neuroretina. Paraffin retinal cross sections derived from C57BL/6J mice, (**A**–**C**,**E**,**F**, n = 12) and PAR1^−/−^ mice (n = 2) (**D**) were stained with anti-PAR1 antibody (red, **A**,**C**,**D**), secondary antibody only, (**E**) and commercial mouse IgG (**F**) as controls and were counter-stained with 4′,6-diamidino-2-phenylindole (DAPI, blue, **B**,**D**-insert). Insert in panel D: the insert demonstrates DAPI staining of this section. Scale bars: 50 μm. GCL: ganglion cell layer, IPL: inner plexiform layer, INL: inner nuclear layer; OPL: outer plexiform layer, ONL: outer nuclear layer, OS: (photoreceptor) outer segments.

**Figure 2 ijms-23-00984-f002:**
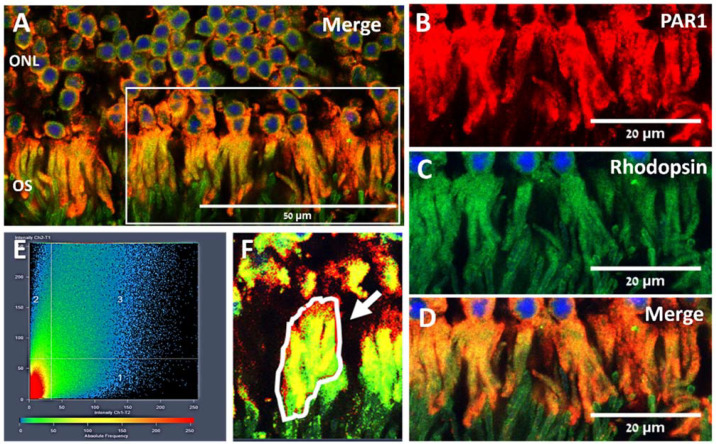
PAR1 co-localization with rhodopsin in mouse retina. A representative image of paraffin retinal sections from C57BL/6J mice co-stained with anti-PAR1 (red) and anti-rhodopsin (green) antibodies. Nuclei were counter-stained with DAPI (blue, **A**,**C**,**D**). (**B**–**D**) A larger magnification of the area defined by the white rectangle, showing the red (PAR1, **B**), green and blue (rhodopsin and DAPI, **C**) channels and the merged image (**D**). Images were obtained with a confocal microscope (LSM700). ONL: outer nuclear layer, OS: (photoreceptor) outer segments. (**E**,**F**) Co-localization analysis using ZEN software (LSM software ZEN 2012 SP1 Version 8.1, Zeiss, Jena, Germany), calculating Pearson’s correlation coefficient between the red (PAR1) and green (rhodopsin) channels was performed on region of interest (ROI) as representatively indicated by the white arrow (**F**). Co-localization analysis was performed in three areas of retinal sections derived from two mice.

**Figure 3 ijms-23-00984-f003:**
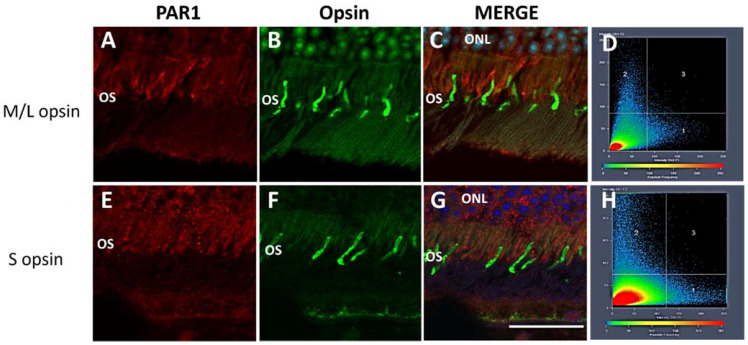
PAR1 does not co-localize with cone L/M-opsin and S-opsin in mouse retina. A representative image of paraffin cross sections of retinas derived from C57BL/6J mice stained with anti-PAR1 (red, **A**,**C**,**E**,**G**), L/M opsin (green, **B**,**C**), and S opsin (green, **F**,**G**) antibodies. Nuclei were counter-stained with DAPI (blue, **C**,**G**). Images were obtained with a confocal microscope (LSM700). ONL: outer nuclear layer, OS: (photoreceptor) outer segments. (**D**,**H**) Co-localization analysis was performed in three areas of retinal sections derived from two mice (n = 2). Scale bar: 25 μm.

**Figure 4 ijms-23-00984-f004:**
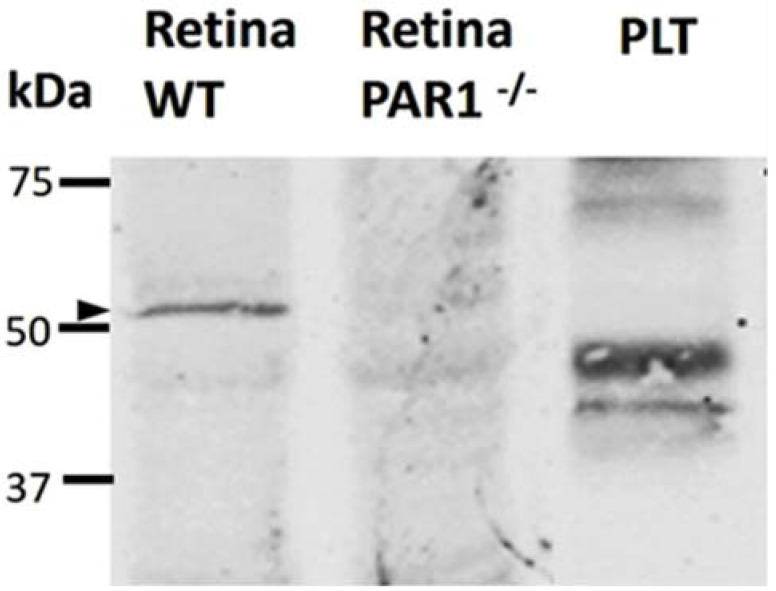
Western blot analysis with PAR1 antibody confirmed the specificity of antibodies. Western blot analysis was performed with the same antibody used for immunofluorescence analysis (NBP-71770, Novus biologicals). A ~52 kDa protein was detected (indicated by an arrow) in lysates of retinas derived from C57BL/6J mice but not from PAR1 knockout mice (PAR1^−/−^) nor in mouse platelets (PLT). The uncropped membrane is presented in Appendix A.

**Figure 5 ijms-23-00984-f005:**
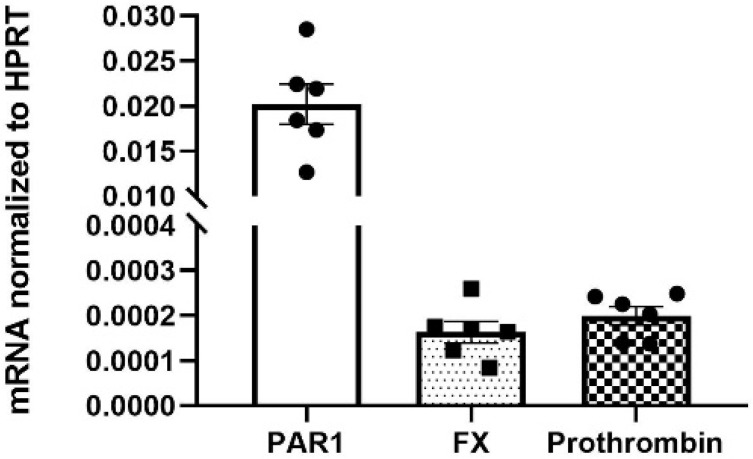
Expression of coagulation factors mRNA in mouse neuroretina. The mRNA expression levels of Factor X (FX), prothrombin and PAR1 in the neuroretina were determined by quantitative real-time reverse transcriptase PCT (qRT-PCR) analysis of neuroretinas derived from six mice (13-week-old). Results are presented relative to hypoxanthine guanine phosphoribosyltransferase expression using the 2^ΔCT calculating method.

**Figure 6 ijms-23-00984-f006:**
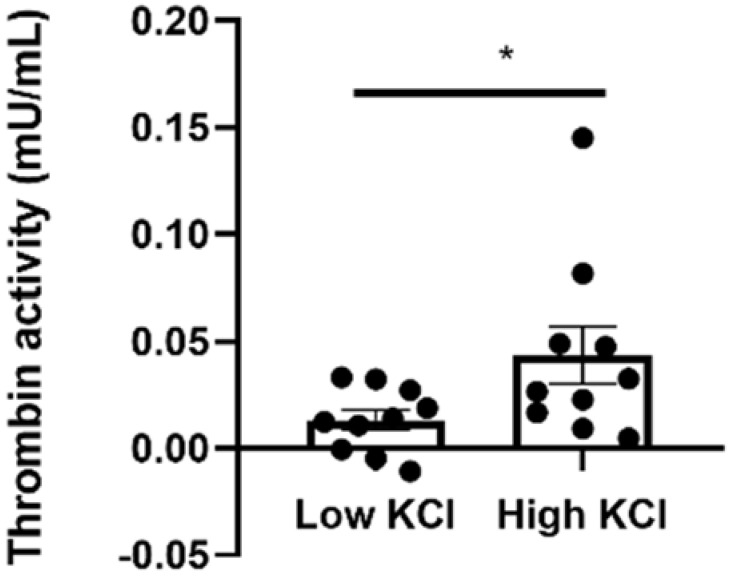
Thrombin activity in isolated neuroretinas under low and high KCl concentrations ex vivo. Thrombin activity was measured in neuroretinas derived from C57BL/6J mice incubated ex vivo in a buffer containing high (56 mM, n = 10) or low (5.6 mM, n = 10) KCl concentration. Significant higher thrombin activity was measured in high KCl compared to the control in low KCl conditions. * *p* < 0.05.

## Data Availability

All data are available upon reasonable request.

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
