# Peer review of "Intrinsic Expression of Coagulation Factors and Protease Activated Receptor 1 (PAR1) in Photoreceptors and Inner Retinal Layers"

_ijms, 2022, doi:10.3390/ijms23020984_

Round 1
Reviewer 1 Report
This study is to characterize the distribution of the thrombin receptor, Protease Activated Receptor 1 (PAR1), in the neuroretina. The results indicate that the PAR1 is expressed in the outer neuroretina in rod photoreceptors but not cones, as well as in the inner nuclear layer, and demonstrate an elevation in thrombin activity in isolated neuroretina in response to KCl depolarization ex-vivo. It may be accepted for publication after minor revisions.
1, The Figure numbers in the figure legends are missing, please checked and modified. Figure 6 is not necessary. The contrast ratio of Figure 4 can be improved, the quality (resolution) of all the figures can be further improved.
2, Please carefully check the whole manuscript. For example, line 82, “PARNo”; line 235, “13-week-old male”, it is better not to start a sentence with numeral; The unit “ml” can be changed to “mL”; There should be a space between the numeral and units.
3, A conclusion section can be added at the end of the manuscript.
Author Response
Dear Editor and Reviewers,
Thank you for your thorough and positive review of the manuscript titled “Intrinsic Expression of Coagulation Factors and Protease Activated Receptor 1 (PAR1) in Photoreceptors and Inner Retinal Layers”. Please consider our revised (IJMS-1542421) for publication in International Journal of Molecular Sciences in the Molecular Neurobiology section.
We appreciate your feedback and have thoroughly reviewed the manuscript edited the text, performed additional experiments, and added more results accordingly. Below is our point-by -point response to each reviewer in blue. Our changes in the text are written in blue. We believe that this study emphasizes the specific localization of the intrinsic functional PAR1 pathway in the retina. We hope that the reviewers’ comments have been addressed and we have edited the manuscript accordingly. We now feel that the manuscript is ready for publication.
We would like to take this opportunity to thank you and the reviewers for the constructive comments and positive attitude. We truly believe that the reviewers' observations have substantially improved the manuscript.
We look forward to hearing from you.
Sincerely yours,
Efrat Stein-Shavit
Reviewer 1
This study is to characterize the distribution of the thrombin receptor, Protease Activated Receptor 1 (PAR1), in the neuroretina. The results indicate that the PAR1 is expressed in the outer neuroretina in rod photoreceptors but not cones, as well as in the inner nuclear layer, and demonstrate an elevation in thrombin activity in isolated neuroretina in response to KCl depolarization ex-vivo. It may be accepted for publication after minor revisions.
1, The Figure numbers in the figure legends are missing, please checked and modified. Figure 6 is not necessary. The contrast ratio of Figure 4 can be improved, the quality (resolution) of all the figures can be further improved.
We checked and corrected the Figures numbers and thank the reviewer for pointing this out. Figure 6 demonstrates elevated thrombin activity following KCl-induced depolarization in isolated mouse neuroretinas. We believe that this Figure is highly important because it indicates a possible functional role of PAR1/thrombin pathway during neuronal activity in the retina. The contrast of Figure 4 was improved, and the resolution of all Figures was adjusted to 330 DPI.
2, Please carefully check the whole manuscript. For example, line 82, “PARNo”; line 235, “13-week-old male”, it is better not to start a sentence with numeral; The unit “ml” can be changed to “mL”; There should be a space between the numeral and units.
We thank the reviewer for noticing these typos. We have extensively examined the manuscript and corrected them. PARNo was corrected to PAR1 (line 83), the “13-week-old male” sentence was rephrased (page7, line244). The unit “ml” was changed to “mL” and spaces were inserted between the numeral and units (along the manuscript).
3, A conclusion section can be added at the end of the manuscript.
A conclusion section was added at the end of the manuscript (page7, lines 236-241)

Reviewer 2 Report
In this manuscript Goldberg and colleagues describe the expression of PAR1, prothrombin and FX in the mouse neuroretina. The manuscript is clear and, in the complex, well organized but there are some points that need to be addressed.
Major points:
- Line 225, “we hypothesize that other blood cells and retinal endothelial cells are negligible relative to all neuroretinal cells”.
Could you comment and specify why do you think other blood cells and endothelial cells are negligible?
- In the thrombin activity assay, how can you be sure that the two halves of the neuroretina are equal? Have you performed any control?
- You say “In the peripheral nerve system, exposing rat sciatic nerve to high thrombin levels led to a nerve conduction block”. Actually, the effect of PAR1 activation in peripheral nerve appears quite complex. You should better explain this point possibly citing recent review on this matter (Pompili et al., Biomolecules 2021).
- Figure 1E shows that the secondary antibody alone gives some background especially in the OS- (photoreceptor) outer segment. You are using a mouse primary antibody on a mouse tissue, so a certain amount of non-specific staining is expected. Since you are concentrating your attention on the photoreceptor layer, please show the negative control stainings in Figure 3 and 4. Negative controls should be performed using Igs of the same species and the same concentration of the primary antibody.
- In Figure 1 you highlight ONL, IS, OS layers; why in Figure 2 have you reported just ONL and POS layers? Does OS correspond to POS? And where is the IS layer in Figure 2?
Minor points:
- Line 31 sounds as a repetition of line 27
- Line 43 remove “in”
- Remove the dash from “in-vivo”, “in-vitro”, “ex-vivo”
- Line 82, PARNo??
- Line 88, Figure is Figure 1. What are the white bars for?? Even the other figures need to be numbered.
- Correct Novus Biologicals; in text it is often Nuvos Biologicals
- Line 178, “supports”
- Line 179, find a synonym for “supported”
- Line 214, correct Muller
- Line 220, “nervous” instead of “nerve”
- Line 249, Triton-X
- Line 250, SNext ??
- Line 251 SSec ??
- Line 287 “Tris HCl 50mM pH 7.6, NaCl 150mM, Tween 0.1%” should be “50mM Tris HCl pH 7.6, 150mM NaCl, 0.1% Tween 20”
- The PAR1 -/- mice are from Envigo as well?
Author Response
Dear Editor and Reviewers,
Thank you for your thorough and positive review of the manuscript titled “Intrinsic Expression of Coagulation Factors and Protease Activated Receptor 1 (PAR1) in Photoreceptors and Inner Retinal Layers”. Please consider our revised (IJMS-1542421) for publication in International Journal of Molecular Sciences in the Molecular Neurobiology section.
We appreciate your feedback and have thoroughly reviewed the manuscript edited the text, performed additional experiments, and added more results accordingly. Below is our point-by -point response to each reviewer in blue. Our changes in the text are written in blue. We believe that this study emphasizes the specific localization of the intrinsic functional PAR1 pathway in the retina. We hope that the reviewers’ comments have been addressed and we have edited the manuscript accordingly. We now feel that the manuscript is ready for publication.
We would like to take this opportunity to thank you and the reviewers for the constructive comments and positive attitude. We truly believe that the reviewers' observations have substantially improved the manuscript.
We look forward to hearing from you.
Sincerely yours,
Efrat Stein-Shavit
Reviewer 2
In this manuscript Goldberg and colleagues describe the expression of PAR1, prothrombin and FX in the mouse neuroretina. The manuscript is clear and, in the complex, well organized but there are some points that need to be addressed.
Major points:
Line 225, “we hypothesize that other blood cells and retinal endothelial cells are negligible relative to all neuroretinal cells”.
Could you comment and specify why do you think other blood cells and endothelial cells are negligible?
As described in the discussion (page 7, lines 225-229), Chavkin, et al reported that endothelial cells constitute only ~0.6% of all retinal cells [reference 38]. In addition, the choroid was removed before tissue lysis. Since mice platelets lack PAR1 [reference 19], we hypothesize that other blood cells and retinal endothelial cells are negligible relative to all neuroretinal cells.
In the thrombin activity assay, how can you be sure that the two halves of the neuroretina are equal? Have you performed any control?
As indicated in the revised “Methods” section (page 9, lines 322-323, 340-343), isolated neuroretinas were divided into two halves, symmetrically through the optic nerve to ensure equal division. The experiment was controlled by weighing the retinas halves. No significant differences were found between the two half’s of the retina used in both groups (p=0.88).
You say “In the peripheral nerve system, exposing rat sciatic nerve to high thrombin levels led to a nerve conduction block”. Actually, the effect of PAR1 activation in peripheral nerve appears quite complex. You should better explain this point possibly citing recent review on this matter (Pompili et al., Biomolecules 2021).
We thank the reviewer for this important comment. Indeed, the glial PAR1 role in the peripheral nerve axon function and repair is very complex and relevant for this current study rational. We added this to both Introduction (page 2, line 47) and the Discussion (page 7, line 223) sections.
Figure 1E shows that the secondary antibody alone gives some background especially in the OS- (photoreceptor) outer segment. You are using a mouse primary antibody on a mouse tissue, so a certain amount of non-specific staining is expected. Since you are concentrating your attention on the photoreceptor layer, please show the negative control stainings in Figure 3 and 4. Negative controls should be performed using Igs of the same species and the same concentration of the primary antibody.
We have conducted additional experiment using commercial mouse IgG as was suggested. This data was added to Figures 1F and presents negative control for immunofluorescence staining, (Figure 1F, page 3). We believe that this control staining together with the secondary only staining (Figure 1E), as well as lack of positive staining in neuroretinal derived from PAR1-/- mice (Figure 1D) strongly support the specificity of PAR1 staining in the neuroretina.
In Figure 1 you highlight ONL, IS, OS layers; why in Figure 2 have you reported just ONL and POS layers? Does OS correspond to POS? And where is the IS layer in Figure 2?
We have clarified this issue and now the OS is marked in the revised Figures 1,2 and 3 and it describes the photoreceptors outer segments.
Minor points:
Line 31 sounds as a repetition of line 27
The abstract was corrected, and this repetition was removed.
Line 43 remove “in”
Done
Remove the dash from “in-vivo”, “in-vitro”, “ex-vivo”
Done and was corrected along the manuscript
Line 82, PARNo??
Corrected
Line 88, Figure is Figure 1. What are the white bars for?? Even the other figures need to be numbered.
We apologize for the problem in Figure numbers. In the PDF built by the journal website during submission, all figure number appeared correctly. We hope you can see it in the revised manuscript.
The legend to Figure 1 was revised to indicate that the white bars indicate a scale bar of 50 mm.
Correct Novus Biologicals; in text it is often Nuvos Biologicals
Corrected (pages 4, 5, 6, 8 lines 129, 143, 205, 300)
Line 178, “supports”
Corrected
Line 179, find a synonym for “supported”
The sentence was revised: A functional role of this pathway is suggested by the KCl depolarization experiment, indicating an increased thrombin activity in the neuroretinal following depolarization.
Line 214, correct Muller
Corrected
Line 220, “nervous” instead of “nerve”
Corrected
Line 249, Triton-X
Corrected
Line 250, SNext ??
Corrected
Line 251 SSec ??
Corrected
Line 287 “Tris HCl 50mM pH 7.6, NaCl 150mM, Tween 0.1%” should be “50mM Tris HCl pH 7.6, 150mM NaCl, 0.1% Tween 20”
Corrected
The PAR1 -/- mice are from Envigo as well?
As indicated in the revised manuscript (page 7 lines 245-247), the PAR1-/- mice were a generous gift of Prof. Yair Reisner, the Weizmann Institute of Science, Rehovot, Israel.

Round 2
Reviewer 2 Report
I think that now the manuscript fulfils the requirments for publication in IJMS.
Just correct "limits" on lane 47